# QUASAR: QUANTUM ASSEMBLY CODE GENERATION USING TOOL-AUGMENTED LLMS VIA AGENTIC RL

## ABSTRACT

Designing and optimizing task-specific quantum circuits are crucial to leverage the advantage of quantum computing. Recent large language model (LLM)-based quantum circuit generation has emerged as a promising automatic solution. However, the fundamental challenges remain unaddressed: (i) parameterized quantum gates require precise numerical values for optimal performance, which also depend on multiple aspects, including the number of quantum gates, their parameters, and the layout/depth of the circuits. (ii) LLMs often generate low-quality or incorrect quantum circuits due to the lack of quantum domain-specific knowledge. We propose **QUASAR**, an agentic reinforcement learning (RL) framework for quantum circuits generation and optimization based on tool-augmented LLMs. To align the LLM with quantum-specific knowledge and improve the generated quantum circuits, QUASAR designs (i) a quantum circuit verification approach with external quantum simulators and (ii) a sophisticated hierarchical reward mechanism in RL training. Extensive evaluation shows improvements in both syntax and semantic performance of the generated quantum circuits. When augmenting a 4B LLM, QUASAR has achieved the validity of 99.31% in Pass@1 and 100% in Pass@10, outperforming industrial LLMs of GPT-4o, GPT-5 and DeepSeek-V3 and several supervised-fine-tuning (SFT)-only and RL-only baselines.

## 1 INTRODUCTION

Quantum hardware has improved remarkably in recent years (AI & Collaborators, 2025; Bravyi et al., 2024; Bluvstein et al., 2024) and this rapid hardware development creates demand for improved quantum software and algorithms. Quantum software and algorithms can be categorized into classical platforms that support quantum computers themselves, including quantum error mitigation software and quantum compilers. The second category comprises domain-specific quantum algorithms, including examples like Shor's algorithm and Grover's algorithm. At the core of quantum software and algorithms is the quantum circuit model (Nielsen & Chuang, 2010), which is an assembly-level abstraction for operating gate-based quantum computers. Most of the quantum algorithms can be expressed as quantum circuits (Jordan, 2025).

The desigin of quantum circuits is the foundation in quantum compilers and quantum algorithm development. In this paper, we consider quantum assembly code, i.e., *Open Quantum Assembly Language* (OpenQASM) (Cross et al., 2022), to represent and model quantum circuits due to its generality and machine-independence – the generated circuits can be deployed on any quantum machines without binding to specific vendors. Unlike Python-based quantum programming languages (e.g., Qiskit (Javadi-Abhari et al., 2024a) and Cirq (Developers, 2025)), OpenQASM is a low-level language and closer to the QPU hardware (similar to classical assembly languages in CPUs). The value of OpenQASM lies in platform-agnostic quantum software stacks, including quantum software-hardware co-design, performance characterization, and cross-platform benchmarking. Leading quantum hardware vendors offer OpenQASM as an interface to their QPUs, alongside their own development kits (e.g., Braket SDK (Amazon Web Services), Cirq and Qiskit).

Despite their central role in quantum computing, quantum circuits present two major difficulties in practice. First, they form a complex abstraction to define quantum algorithms compared to classical methods, making them difficult even for expert practitioners (Haferkamp et al., 2022). Second, such

difficulties are further amplified in OpenQASM code, which is particularly error-prone to write for complex quantum algorithms Cross et al. (2022) due to its unique low-level grammar and syntax.

We identify three key challenges in quantum circuit generation: **(1)** QASM code includes numerous numerical parameters in parametrized gates, which are difficult for LLMs to handle accurately; **(2)** unlike classical code generation, QASM evaluation is nontrivial, as its correctness depends not only on syntactic validity but also on the underlying quantum semantics, which are inherently probabilistic and nondeterministic; and **(3)** LLM-generated QASM can fail in various ways (see Figure 1), including compilation error, producing incorrect qubit counts and low-quality parameters.

---

**Hamiltonian Path Problem (QAOA Optimization) :**
Design a QASM 3.0 quantum circuit with 1 qubits and 3 layers to solve the hamiltonian_path starting from node 0 and ending at node 2. given the graph: {"directed": false, "multigraph": false, "graph": {}, "nodes": [{"id": 0}, {"id": 1}, {"id": 2}], "edges": [{"weight": 7, "source": 0, "target": 1}, {"weight": 14, "source": 0, "target": 2}, {"weight": 7, "source": 1, "target": 2}]}.
Provide valid QASM 3.0 code with optimal parameters.

---

| (a) OPENQASM 3.0; | (b) OPENQASM 3.0; | (c) OPENQASM 3.0; | (d) OPENQASM 3.0; |
|---|---|---|---|
| include "stdgates.inc"; | include "stdgates.inc"; | include "stdgates.inc"; | include "stdgates.inc"; |
| bit[1] c; | bit[1] c; | bit[1] c; | bit[1] c; |
| qubit[1] q; | qubit[2] q; 👎 | qubit[1] q; | qubit[1] q; 👍 |
| h q[0]; | h q[0]; | h q[0]; | h q[0]; |
| rz(-0.0924) q[0]; | rz(-0.0924) q[0]; | rz(-0.3324) q[0]; | rz(-0.0924) q[0]; |
| h q[0]; | h q[0]; | h q[0]; | h q[0]; |
| rz(-0.0644) q[0]; | rz(-0.0644) q[0]; | rz(-0.9456) q[0]; | rz(-0.0644) q[0]; |
| h q[0]; | h q[0]; | h q[0]; | h q[0]; |
| rz(-0.1062) q[0]; | rz(-0.1062) q[0]; | rz(-0.7194) q[0]; 👎 | rz(-0.1062) q[0]; |
| h q[0]; | h q[0]; | h q[0]; | h q[0]; |
| rz(-0.0886) q[0]; | rz(-0.0886) q[0]; | rz(-0.2783) q[0]; | rz(-0.0886) q[0]; |
| h q[0]; | h q[0]; | h q[0]; | h q[0]; |
| rz(-0.0668) q[0]; | rz(-0.0668) q[0]; | rz(-0.3762) q[0]; | rz(-0.0668) q[0]; |
| h q[0]; | h q[0]; | h q[0]; | h q[0]; |
| rz(-0.0676); 👎 | rz(-0.0676) q[0]; | rz(-0.8357) q[0]; | rz(-0.0676) q[0]; |
| h q[0]; | h q[0]; | h q[0]; | h q[0]; |
| c[0] = measure q[0]; | c[0] = measure q[0]; | c[0] = measure q[0]; | c[0] = measure q[0]; |

**Fig. 1: Illustration of four possible outcomes for generated OpenQASM code: (a) fails to compile; (b) compiles but uses an incorrect number of qubits; (c) compiles with the correct number of qubits but suboptimal parameters; (d) the desired case — compiles successfully, uses the correct number of qubits, and achieves near-optimal parameters.**

we present QUASAR, an agentic RL post-training framework for LLMs to generate quantum circuits in OpenQASM 3.0 QUASAR achieves this through two key innovations. First, to equip the model with a quantitative understanding of parameterized gates, we develop an agentic RL pipeline with a carefully designed external quantum verification tool, in a LLM interacts directly with quantum simulators. Second, we design a hierarchical four-level reward mechanism that enforces correctness as a prerequisite and, conditioned on correctness, promotes stronger task alignment in the generated circuits. While LLMs are not considered native optimizers for quantum circuit parameters, QUASAR shows that they can learn beneficial ansatz patterns and initial parameter configurations for quantum optimization problems, as illustrated in Figure 2.

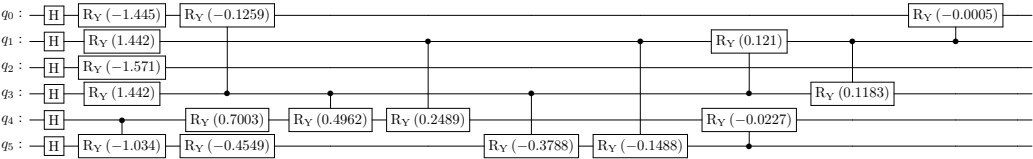

**Fig. 2: Example of an LLM-generated ansatz with initial parameters.**

More precisely, the hierarchical reward mechanism consists of four stages. **(1)** The syntax reward considers whether the LLM-generated OpenQASM code can be parsed. If this fails, the other rewards cannot be computed, and thus, this reward is set relatively high. **(2)** The system computes Jensen–Shannon entropy between the LLM-generated and ground truth distributions in computational basis states, which creates a so-called distributional alignment term for the model.

**(3)** The third reward is computed by considering the expectation value discrepancies between LLM-generated and ground-truth circuits to capture task-specific alignment. This becomes task-specific, since computing expectation values requires the usage of the problem-specific cost Hamiltonian. **(4)** The fourth part of the reward function considers the usability of the LLM-generated circuit in a realistic setup where the circuit optimization continues. This reward counts the number of optimization steps and computes the final optimized expectation value. The final reward for the fourth part is a weighted sum of these two elements.

In this paper, we make the following contributions.

- We design an end-to-end processing pipeline to combine supervised fine-tuning and RL-based post-training, which efficiently interacts with an external quantum verification tool.

- We design a 4-level hierarchical reward mechanism for RL that jointly optimizes syntactic validity, distributional similarity, expectation value differences, and the number of required optimization steps to generate quantum circuits tailored for specific optimization problems.

- We evaluate QUASAR to augment an example LLM, a 4B-Qwen3 model, which outperforms several leading industrial LLMs. The results demonstrate that LLMs can generate practical ansatz patterns and parameter initializations for *Quantum Approximate Optimization Algorithm* (QAOA) and *Variational Quantum Eigensolver* (VQE) quantum circuits, highlighting QUASAR's potential in scalable quantum circuits and algorithm design.

## 2 RELATED WORKS

In recent years, there has been increasing interest in applying GPT- and LLM-based tools to address challenges in designing and constructing circuits for various quantum computational challenges. In one of the first works, IBM Quantum (2025) fine-tuned large language models to serve as a Qiskit Code Assistant, which was evaluated with (Vishwakarma et al., 2024). This work was further improved with post-training reinforcement learning (Dupuis et al., 2025a). Campbell et al. (2025) leveraged Chain-of-Thought (CoT) reasoning and Retrieval-Augmented Generation (RAG) in a multi-agent setting to facilitate the synthesis of quantum circuits. These efforts are closely tied to the Qiskit framework, whereas our approach targets OpenQASM 3.0 Cross et al. (2022), a platform-independent standard now supported by Qiskit, PennyLane, and Cirq. Large language models have also been used for optimizing ansatz design (Liang et al., 2023; Ueda & Matsuo, 2025), and Arlt et al. (2024) used language models to design quantum experiments. Gujju et al. (2025) utilized LLMs for guided ansatz design in financial modeling.

Compared to LLM-based methods, standard transformers and GPT-based systems have received more attention. Fitzek et al. (2024) trained the standard GPT model to predict measurement outcomes from a neutral atom quantum computer. The findings revealed limitations in the standard GPT model's ability to predict measurement outcomes, which could prove valuable in expanding our knowledge of the boundaries of present-day LLMs. Using the standard transformer-based approach, Nvidia has designed an optimization pipeline that produces quantum circuits, which are aimed at identifying ground states in electronic structure Hamiltonians (Nakaji et al., 2024). Since the trainable parameters are in the transformer model, the method tries to circumvent some of the problems in current variational methods, like barren plateaus. Apak et al. (2024) developed Ket-GPT, which uses a GPT-based model to generate realistic quantum circuits. The model is trained in QASMBench circuits (Li et al., 2023). The circuits are restricted to the OpenQASM 2.0 format, without supporting parameters. Finally, Tyagin et al. (2025) developed QAOA-GPT, which is capable of generating QAOA-type circuits that solve the MaxCut problem. Dupuis et al. (2025b) is the first to use a quantum-verifiable reward to post-training LLMs to generate Qiskit code.

## 3 PRELIMINARIES

### 3.1 QUANTUM COMPUTING AND QUANTUM OPTIMIZATION

Quantum computing exploits the principles of quantum mechanics to perform computations using quantum bits (qubits) instead of classical bits. A qubit is a unit vector in a two-dimensional Hilbert

space and can exist in a superposition $\alpha|0\rangle + \beta|1\rangle$ with $\alpha, \beta \in \mathbb{C}$ and $|\alpha|^2 + |\beta|^2 = 1$ (Nielsen & Chuang, 2010). Multi-qubit states arise via tensor products, and computation proceeds by applying unitary gates. Rotation gates $R_x(\theta)$, $R_y(\theta)$, $R_z(\theta)$ are unitary for all $\theta$, and together with entangling operations (e.g., CNOT) and standard gates (e.g., Hadamard, CZ), they form a universal gate set (Yang et al., 2023). By combining these gates into parameterized circuits $U(\theta)$, one obtains a flexible framework for variational quantum algorithms.

One of the most promising applications is quantum optimization (Abbas et al., 2024). Classical problems such as QUBO and HUBO (Lucas, 2014; Boros & Hammer, 2002) can be rewritten in terms of spin variables and mapped to Pauli-Z operators, turning them into Hamiltonian minimization tasks. The goal is to prepare a parameterized state $U(\theta)|0\rangle$ whose expectation value $\langle 0|U^\dagger(\theta)HU(\theta)|0\rangle$ approximates the ground state of $H$. Hybrid quantum-classical methods iteratively optimize $\theta$: QAOA uses a cost Hamiltonian and a mixer ansatz (Farhi et al., 2014), VQE employs expressive or hardware-efficient ansatzes (Peruzzo et al., 2014), and adaptive VQE constructs circuits gate by gate from a predefined pool (Grimsley et al., 2019).

## 3.2 OPENQASM LANGUAGE

**OpenQASM (Open Quantum Assembly Language)** (Cross et al., 2022) is a low-level programming language designed for expressing quantum circuits and operations, which is similar to traditional Hardware Description Language (HDL) like Verilog and VHDL. It serves as an intermediate representation (IR) for quantum algorithms, allowing them to be executed on various quantum hardware platforms. OpenQASM provides a standardized way to describe quantum gates, measurements, and other operations, making it easier for developers to write and share quantum programs.

OpenQASM is widely used in the quantum computing community, since many popular quantum computing software frameworks, such as IBM's Qiskit (Javadi-Abhari et al., 2024b), Google's Cirq (Developers, 2025), Microsoft's QDK (Microsoft), and Rigetti's Forest(Smith et al., 2017), support OpenQASM as a means of serializing quantum circuits in a standard way. This allows developers to write quantum algorithms in a high-level language and then compile them down to Open-QASM for distribution between different quantum hardware backends. OpenQASM has become the conjunction between quantum software and hardware, being critical for both platform-agnostic and platform-specific optimization, mapping, scheduling, evaluation, profiling, and simulation.

## 3.3 AGENTIC REINFORCEMENT LEARNING WITH TOOL USE

Agentic Reinforcement Learning with Tool use (ARLT) can efficiently enhance domain-specific performance in the LLM post-training stage, allowing LLMs to use tools to interact with the environment and learn from verified feedback. Specifically, the policy LLM $\pi_\theta$ aims to predict the next token $\tau_t$ based on the context $\tau_{<t}$. The entire response $\tau$ will be evaluated by the verified reward function $R$. GRPO (Shao et al., 2024) is the commonly used RL algorithm in recent ARLT works (Jiang et al., 2025; Qian et al., 2025), and has been proven successful since DeepSeek-AI et al. (2025). Specifically, given a group of rollout trajectories $\{R(\tau_i)\}_{i=1}^G$, the objective of GRPO is defined as follows:

$$\mathcal{J}(\theta) = \frac{1}{G} \sum_{i=1}^{G} \frac{1}{|\tau^i|} \sum_{t=1}^{|\tau^i|} \min\left[r_t^i(\theta) \cdot \hat{A}_t^i, \text{clip}\left(r_t^i(\theta), 1-\epsilon, 1+\epsilon\right) \cdot \hat{A}_t^i\right], \tag{1}$$

where $r_t^i(\theta) = \frac{\pi_\theta(\tau_t^i|\tau_{<t}^i)}{\pi_{\text{old}}(\tau_t^i|\tau_{<t}^i)}$ is the token-level importance ratio and $\hat{A}_t^i = \frac{R(\tau^i) - \text{mean}(\{R(\tau^j)\}_{j=1}^G)}{\text{std}(\{R(\tau^j)\}_{j=1}^G)}$ is the normalized advantage.

## 4 QUASAR DESIGN

### 4.1 RL POST-TRAINING PIPELINE

**Online Interaction with a Quantum Agent.** Building on Jiang et al. (2025), our quantum agent consolidates the environment and reward computation into a single tool module (see Figure 3). At each RL step: (A) the language model proposes an OpenQASM circuit and calls the external tool

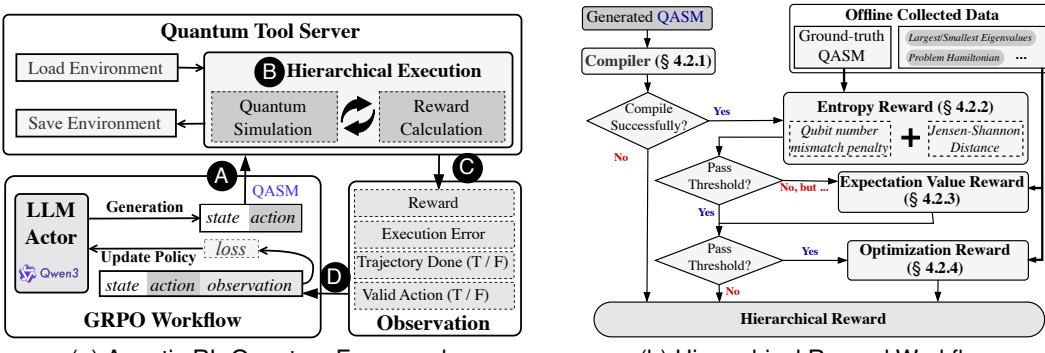

Fig. 3: **QUASAR** design: (a) agentic RL-quantum framework, and (b) hierarchical reward.

– Quantum Tool Server, via HTTP; (B) the agentic tool executes certain quantum simulation and computes a verifiable reward; (C) it returns structured feedback, which includes the scalar reward, execution errors, validity flags, and a trajectory trace; and (D) the training loop ingests this feedback and updates the policy via GRPO. To bootstrap syntactic competence, we first perform supervised fine-tuning without intermediate reasoning traces (non-CoT SFT) using the dataset of Jern et al. (2025). We then apply ARLT to further improve both syntactic and semantic correctness, using simulation-derived signals to shape the policy through policy-gradient updates.

**Quantum Verification.** LLMs can generate code in languages such as Python, C++, and x86-64 based on (Wei et al., 2025), but generating low-level quantum circuits is particularly challenging (Fu et al., 2025). OpenQASM is a domain-specific language with limited exposure during pretraining, making it difficult for models to produce syntactically correct and semantically meaningful circuits. In addition, designing quantum circuits requires a deep understanding of the target optimization problem and underlying quantum computing principles. To address these challenges, we integrate an *agentic quantum verification module* into our RL framework. This module simulates the generated OpenQASM code and evaluates its performance on carefully designed quantum tasks. The resulting metrics are transformed into reward signals that guide the training.

## 4.2 REWARD DESIGN

**Hierarchical reward mechanism.** We introduce a hierarchical reward that covers four key aspects of the generated OpenQASM code: (i) *syntactic correctness*, (ii) *distributional alignment* (an entropy-based term), (iii) an *expectation-value* term, and (iv) an *optimization-progress* term. The computation is hierarchical: we first verify the syntactic correctness of the generated OpenQASM; if it is valid, we then measure the distributional discrepancy between the generated and ground-truth circuits (e.g., via the Jensen–Shannon distance $D_{\mathrm{JS}}(p_{\mathrm{gen}}, p_{\mathrm{gt}})$) to quantify overall mismatch. Notably, generated circuits often realize unitaries $\hat{U}$ that deviate substantially from the ground truth $U^\star$, leading to markedly different measurement distributions in the computational basis. Yet for a given quantum optimization task, performance can still be comparable because evaluation is with respect to the problem Hamiltonian $H$, via $E(\psi) = \langle\psi|H|\psi\rangle$. To capture this task-specific behavior, we augment the objective with two problem-aware terms: (a) an *expectation-value reward* that calculates the distance between the eigenvalues of the generated circuit and the ground truth circuit, and (b) an *optimization-progress reward* that credits the improvement achieved by a local optimizer. The reward is based on the optimization steps from the generated circuit to the optimal, and the gap between the final converged circuit and the ground truth one.

In addition to the above rewards, we introduce a *qubit-mismatch penalty* in stage (ii) to discourage the generation or disappearance of qubits. This penalty addresses a common issue in our training loop, where LLMs often generate circuits with a qubit count inconsistent with the ground-truth circuit, leading to reward calculation errors. For example, a prompt requesting a 9-qubit circuit may result in the LLM producing `qubit[7] q;`.

We adopt a hierarchical procedure to compute rewards, as illustrated in Figure 3b. Specifically, if the distributional divergence between the generated and ground-truth circuits exceeds a predefined threshold, we additionally evaluate a Hamiltonian-based expectation-value reward. If either the (normalized) expectation-value reward surpasses its threshold or the distributional divergence falls below its threshold, we proceed to run a local optimizer and assign an optimization-progress reward. This hierarchical design ensures the objective is robust to circuits that differ in distribution but are equally fit for the given task. The intuition is that a high entropy-based reward already indicates strong distributional alignment, making expensive eigenvalue comparisons unnecessary. Conversely, if the entropy-based reward is low, we assess problem-specific similarity via eigenvalues. Only when either the distributional or expectation-value scores are sufficiently high do we perform local optimization; otherwise, the circuits are deemed too low-quality to justify further evaluation. In the following, we introduce each of the four reward components in detail.

### 4.2.1 SYNTACTIC REWARD

Syntactic reward refers to the syntactic correctness of the circuits. A circuit is considered syntactically correct if the Qiskit QASM 3.0 parser can parse it. This implies that it follows the grammatical rules of the OpenQASM standard (Cross et al., 2022). If the generated QASM fails to compile, the reward calculation process ends and returns $-1$.

### 4.2.2 ENTROPY REWARD

Previous work Jern et al. (2025) evaluated the quality of circuits with respect to relative entropy, i.e., Kullback-Leibler divergence. In this work, we design the reward as the Jensen–Shannon distance, which can be understood as a normalized relative entropy to the interval $[0, 1]$ for the stability of RL training. The Jensen-Shannon distance is implemented as

$$d_{\text{JS}} = (p, q) = \sqrt{\frac{\text{JS}(p \, \| \, q)}{\log 2}},$$  (2)

where

$$\text{JS}(p \, \| \, q) = \tfrac{1}{2} D_{\text{KL}}(p \, \| \, m) + \tfrac{1}{2} D_{\text{KL}}(q \, \| \, m),$$

for probability distributions $p$ and $q$ and $D_{\text{KL}}$ is the standard KL-divergence. The distribution $m = \tfrac{1}{2}(p+q)$. The reward is $1 - d_{\text{JS}}(p, q) \in [0, 1]$, so that a reward of $1.0$ is returned for those distributions that are identically the same, and $0$ for those that are very different.

As mentioned earlier, the generated QASM may have a different number of qubits than the ground-truth QASM, which makes the entropy-based reward inapplicable due to the resulting dimension mismatch. Let $n_{\text{gen}}$ and $n_{\text{gt}}$ be the qubit counts of generated and ground-truth QASM and $k = \min(n_{\text{gen}}, n_{\text{gt}})$. We define a qubit-mismatch penalty

$$R_{\text{qm}} = \text{clip}_{[-0.2, 0]}\big(\alpha + \beta \, \Delta n + \gamma \, a_{\text{extra}} + \eta \, e_{\text{cross}}\big), \quad \Delta n = |n_{\text{gen}} - n_{\text{gt}}|,$$

where $a_{\text{extra}}$ counts *active* extra qubits beyond the first $k$ (idle or reset-only ancillas incur little or no penalty) and $e_{\text{cross}}$ counts multi-qubit gates that entangle extras with core wires. To ensure fairness, the distribution term $D_{\text{JS}}$ is computed on the first $k$ qubits (marginalization), and its weight is down-scaled by the mismatch severity.

Note that for the expectation-value term and the optimization-progress term below, we pad the problem Hamiltonian with identities to ensure that its width matches the circuit, thereby preserving comparability while preventing reward hacking via ancillary qubits.

### 4.2.3 EXPECTATION-VALUE REWARD

The training dataset consists of optimization problems expressed as eigenvalue minimization problems. Measuring from an ideal circuit would return the optimal eigenvalue with probability $1$. This means that the expectation value from this circuit would coincide with the eigenvalue. In the realistic cases, the measured expectation values over the weighted eigenvalues from the circuits are always larger than the optimal result. The expectation value from the LLM-generated circuit and the ground truth optimal eigenvalue enable us to construct the third reward function as follows.

For a syntactically correct LLM-generated quantum circuit, we simulate the circuit and compute the expectation value of the problem-specific cost Hamiltonian. Let his value be $E_{\text{gen}}$. To establish a reference for comparison, we also compute the minimum (optimal solution) and maximum eigenvalues for the problem-specific cost Hamiltonian. Let these values be $E_{\text{min}}$ and $E_{\text{max}}$.

Because $0 \leq E_{\text{min}} < E_{\text{gen}} < E_{\text{max}}$, we calculate the min-max normalized value as

$$f_{\text{max}}^{\text{min}}(E_{\text{gen}}) := \frac{E_{\text{gen}} - E_{\text{min}}}{E_{\text{max}} - E_{\text{min}}}, \tag{3}$$

which is again a value in the interval $[0, 1]$.

### 4.2.4 Optimization Reward

It is unlikely that LLMs generate parameterized quantum circuits that are immediately solutions to the given optimization problems. Thus, the realistic pipeline includes a phase where the user continues optimizing the parameters in the LLM-generated circuit. Given the complexity of the parameter optimization problem, the most realistic measure of the usefulness of the LLM-generated circuit is the number of optimization steps required to achieve an optimized circuit, where a sufficiently low expectation value can be measured. Hence, the system implements a module that allows for ongoing optimization. In this case, the reward is defined as $1/(1+n)$, where $n$ is the number of optimization steps required for the LLM-generated circuit to reach an optimized quantum circuit for the given optimization problem. Additionally, the system should favor those generated quantum circuits with which the lower expectation value can be obtained. Let $E_{\text{opt}}$ be the expectation value after the optimization loop has been applied to the LLM-generated quantum circuit. Let $E_{\text{min}}$ again be the ideal ground truth, i.e., the optimal eigenvalue. Then, we include

$$\frac{1}{1+n} + f_{\text{max}}^{\text{min}}(E_{\text{opt}}), \tag{4}$$

where $f_{\text{max}}^{\text{min}}$ was defined in Equation 3.

## 5 Experiments

### 5.1 Experimental Setup

**Training Setup.** We utilize the training data from (Jern et al., 2025), which is one of the most extensive available datasets of quantum circuits in QASM format covering 12 central optimization problem primitives on graphs (Karp, 1972), and many of their abstract descriptions appeared in (Lucas, 2014). Since Jern et al. (2025) does not describe these problems in detail, we include a more detailed description for each problem in the Appendix D.2. The key characteristics in this dataset are the parametrized circuits with optimal parameters, the problem Hamiltonians, and the smallest and largest eigenvalues for the Hamiltonians. The full dataset is constructed so that for each optimization problem, the system generates a random graph, where the problem is solved using QAOA, VQE, and adaptive VQE. If the quantum optimization problem is simulable and optimization converges, the problem, graph, circuits in QASM format, and Hamiltonian, along with other data, are included in the training dataset. The complete details of the problem are provided in the Appendix D.2.

We fine-tune a 4B SFT model with GRPO on 16×H100-64GB GPUs using FSDP (Zhao et al., 2023). Each prompt samples $n=16$ rollouts via vLLM (Kwon et al., 2023) (temperature 0.7, top-$p = 0.8$). The average training time is 48 hours. More details can be found in Appendix B.

**Evaluation Metrics.** We evaluate the fine-tuned model across four complementary metrics designed to assess both syntactic fidelity and optimization quality. First, we measure *syntactical correctness ratio (SCR)*, defined as the percentage of generated outputs that can be parsed as valid OpenQASM 3.0 circuits using Qiskit's QASM parser. This metric captures whether the model has internalized the grammar of the domain-specific language. Second, we perform *successful rate of expectation value (SREV)*, where each syntactically correct circuit is simulated and the expectation value for the problem-specific cost Hamiltonian $H$ is computed. We compute the expectation value of generated circuit $\mathcal{C}$ as $E(\mathcal{C}) = \langle \psi_{\mathcal{C}} | H | \psi_{\mathcal{C}} \rangle$. Let $E^{\star}$ denote the ground-truth QASM's expectation. The circuit is counted as *successful* if $|E(\mathcal{C}) - E^{\star}| \leq 0.2$, and SREV is the percentage of successful

QASMs. Third, we evaluate the ***relative entropy (RE) of probability distributions*** by computing the KL divergence $D_{\mathrm{KL}}(P_{\mathrm{sol}} \| P_{\mathrm{gen}})$ between the outcome distribution of the generated circuit and that of the optimized reference. Finally, we compute the ***High-Quality Circuit Ratio (HQCR)***, defined as the proportion of generated circuits whose relative entropy against the ground-truth distributions is within a threshold $0.1$. This metric provides a more interpretable measure of how often the model produces reasonable solutions. Each metric is measured in Pass@1 and Pass@10, where Pass@1 evaluates whether a single sampled QASM per prompt meets the criterion, and Pass@10 evaluates whether at least one out of ten sampled QASMs meets the criterion. Results are shown in Table 1, where up/down arrows by column names indicate whether higher or lower values are better.

**Baselines.** The baselines can be categorized into three groups. **(i)** ***Prompting-Based State-of-the-art LLMs***: We evaluate DeepSeek-V3 (Liu et al., 2024) and OpenAI's GPT-4o (Hurst et al., 2024) and GPT-5 (OpenAI, 2025) via their official API, with GPT-5 being OpenAI's latest flagship. All models are evaluated using few-shot prompting with four demonstration examples. **(ii)** ***SFT-only***: We train Qwen-3-4B using SFT only. **(iii)** ***RL-only***: We evaluate a cold-start model trained using the GRPO algorithm (Shao et al., 2024).

## 5.2 RESULTS

**Tab. 1: QUASAR's Pass@K and comparison with existing techniques.**

| Methods | Pass@1 | | | | Pass@10 | | | |
|---|---|---|---|---|---|---|---|---|
| | SCR ↑ | SREV ↑ | RE ↓ | HQCR ↑ | SCR ↑ | SREV ↑ | RE ↓ | HQCR ↑ |
| Few-Shot Prompting | | | | | | | | |
| DeepSeek-V3 | 94.83% | 12.24% | 19.20 | 10.00% | 98.97% | 26.38% | 16.39 | 16.38% |
| GPT-5 | 87.07% | 10.00% | 19.94 | 6.90% | 90.52% | 27.07% | 11.57 | 16.55% |
| GPT-4o | 87.93% | 9.83% | 19.42 | 6.38% | 88.79% | 18.62% | 14.08 | 12.07% |
| Post-Training (Qwen3-4B) | | | | | | | | |
| SFT | 97.41% | 18.97% | 12.74 | 15.17% | 99.65% | 31.55% | 10.81 | 23.62% |
| Cold Start GRPO | 84.48% | 19.84% | 14.32 | 12.41% | 95.17% | 27.59% | 11.38 | 18.96% |
| **QUASAR (ours)** | **99.31%** | **22.41%** | **11.61** | **17.24%** | **100%** | **33.10%** | **8.48** | **27.24%** |

**Performance Analysis.** We report the performance of QUASAR in Table 1. QUASAR achieves strong results against all baselines, with a pass@1 of $99.31\%$ SCR. Compared to the best competing methods, it yields a $+12.95\%$ improvement in SREV ($22.41$ vs. $19.41$ for Cold Start (GRPO)), an $+8.87\%$ reduction in RE ($11.61$ vs. $12.74$ for SFT), and a $+13.64\%$ gain in HQCR over SFT. For pass@10, we even achieve $100\%$ syntactical correctness and even better semantic improvement.

Meanwhile, as noted earlier, HQCR was measured with a fixed threshold of $0.1$. In Figure 4, we vary this threshold from $0.1$ to $0.9$ and report The fraction of valid QASMs that qualify as high-quality. At best, QUASAR achieves a $+13.65\%$ improvement over SFT, $+72.4\%$ over DeepSeek-V3, and up to $1.65\times$ and $1.50\times$ improvements over GPT-4o and GPT-5, respectively. Furthermore, to quantify the semantic closeness of the generated QASMs to the ground truth, we also measure $\Delta E = |E(\mathcal{C}) - E^\star|$ defined before; note that smaller values indicate higher-quality QASMs. Figure 5 illustrates the distribution of $\Delta E$ for each model, QUASAR achieves substantial improvements over all baselines, including a $4.9\%$ reduction in the median $\Delta E$ and a $9.7\%$ reduction in the upper quartile compared to the second-best SFT.

In addition, we compare QUASAR-generated QASMs with a *random parameter initialization baseline*, a common practice in hybrid quantum-classical algorithms. For each QUASAR circuit, we evaluated JS-divergence and expectation value relative to the Hamiltonian and ground truth, alongside 100 randomized variants with parameters sampled from $(-\pi, \pi]$. The results show that QUASAR consistently outperforms random initialization, achieving lower JS-divergence ($0.79$ vs. $0.95$) and expectation values closer to the optimum ($0.16$ vs. $0.36$). Further details are provided in Appendix E.

## 5.3 ABLATION STUDY OF THE HIERARCHICAL REWARD

We quantify the contribution of each reward term by independently removing each component of the hierarchy reward in section 4. For each variant, we disable exactly one component while keeping

the remainder unchanged; we also report a *Validity-only* sanity baseline, which only returns reward based on whether the generated QASM is valid(1) or not(−1). All runs are initialized from the same SFT checkpoint and trained with matched compute (identical optimizer, schedule, batch size, steps, and decoding settings). The evaluation metrics are the same as presented in section 4, and the results are summarized in Table 2.

**Distributional alignment (RE) is the primary driver of all metrics.** Removing RE causes the largest drop across semantic metrics and, unexpectedly, even harms syntactic correctness. This highlights that coarse-grained alignment of measurement distributions is essential for effective reward shaping. **Expectation value (EV) safeguards hard cases.** Disabling EV reduces SREV as expected and slightly degrades other semantic metrics. When distributional divergence is high, EV provides a task-specific signal that rescues otherwise borderline QASMs. **Optimization progress (Opt) provides incremental gains.** Removing Opt leads to a notable drop in HQCR, with a larger gap at Pass@10 than at Pass@1. This suggests that rewarding fewer optimization steps can also benefit the training. **Qubit mismatch penalty (QMP) ensures stability.** Without QMP, qubit-count inconsistencies increase and reward noise emerges (e.g., evaluation failures or padded comparisons), lowering all metrics. **Validity-only rewards are insufficient.** A reward that enforces only basic validity achieves reasonable SCR but lags considerably on SREV, RE, and HQCR, even performing slightly worse than SFT. This underscores the necessity of semantically informed reward signals.

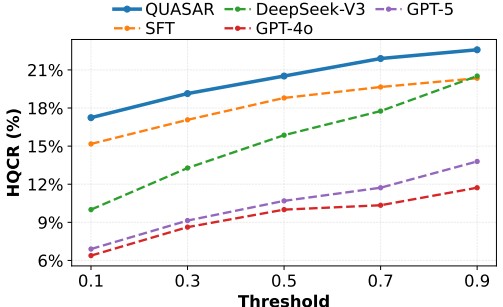

**Fig. 4: HQCR with varying threshold.**

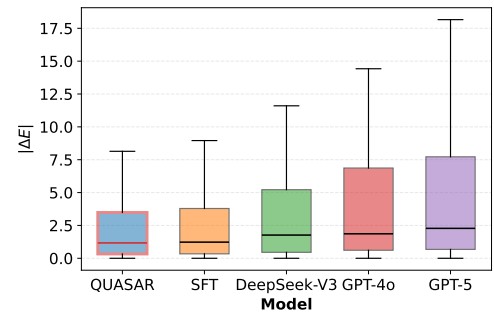

**Fig. 5: $\Delta E$ for valid QASMs.**

**Tab. 2: Reward ablation for QUASAR: contribution of each component.**

| Variant | Components | Pass@1 | | | | Pass@10 | | | |
|---|---|---|---|---|---|---|---|---|---|
| | | SCR ↑ | SREV ↑ | RE ↓ | HQCR ↑ | SCR ↑ | SREV ↑ | RE ↓ | HQCR ↑ |
| **Full (QUASAR)** | **Val + RE + EV + Opt + QMP** | **99.31%** | **22.41%** | **11.61** | **17.24%** | **100%** | **33.10%** | **8.48** | **27.24%** |
| w/o EV term | Val + RE + Opt + QMP | 98.62% | 20.69% | 11.82 | 16.38% | 100% | 31.03% | 9.12 | 26.72% |
| w/o RE term | Val + EV + Opt + QMP | 66.38% | 5.17% | 24.67 | 5.69% | 79.82% | 15.52% | 18.26 | 16.90% |
| w/o Opt term | Val + RE + EV + QMP | 98.79% | 21.90% | 11.98 | 16.90% | 100% | 32.76% | 9.01 | 26.55% |
| w/o QMP | Val + RE + EV + Opt | 98.79% | 21.72% | 12.02 | 16.21% | 100% | 31.55% | 9.48 | 27.06% |
| Validity only | Val | 99.13% | 18.79% | 12.89 | 14.66% | 100% | 30.86% | 11.27 | 23.27% |

# 6 CONCLUSION

We presented QUASAR, an agentic reinforcement learning framework for post-training large language models to generate OpenQASM 3.0 circuits with high syntactic validity and semantic fidelity. By integrating an external verification tool with quantum-aware RL and a hierarchical reward that enforces syntax, aligns distributions, reduces expectation-value errors, and promotes optimization efficiency, QUASAR consistently outperforms leading industrial LLMs and SFT-only and RL-only baselines across quantum optimization benchmarks. Our results highlight that distributional alignment is the key driver of quality, while expectation-value and optimization-progress terms provide complementary gains, demonstrating that tool-augmented RL can effectively bridge general-purpose LLMs and domain-specific quantum code generation, paving the way for broader applications in automated quantum algorithm design. Appendix F discusses the limitations and future directions.

## ETHICS STATEMENT

We affirm that all authors have read and adhere to the ICLR Code of Ethics. This work does not involve human or animal subjects, sensitive personal data, or privacy risks. The use of LLMs was limited to providing writing support and refining language. LLMs were not used in the design of algorithms, the development of theoretical results, or the execution of experiments, ensuring that all core scientific contributions are entirely the work of the authors.

## REPRODUCIBILITY STATEMENT

We provide the implementation details for reproducing our experimental results in Appendix B.

Once the paper is published, we will open source the training and evaluation code on GitHub, and model weights on Hugging Face to public. They can also be uploaded to the anonymous GitHub during the public discussion period when requested.

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

## A  LLM USAGE STATEMENT

In this work, the use of large language models (LLMs) was restricted to writing support and language refinement. Specifically, LLMs assisted in enhancing the clarity and coherence of the manuscript. LLMs were not used in the design of algorithms, the development of theoretical results, or the execution of experiments. All core scientific contributions are entirely the work of the authors.

## B  IMPLEMENTATION DETAILS

Our QUASAR is built on Verl-Tool (Jiang et al., 2025), and we mainly adopted the hyperparameter settings from Verl-Tool. The base model for SFT is Qwen3-4B-Instruct-2507, and we perform SFT adopting the same setting from Jern et al. (2025). Training runs on 16 NVIDIA H100-64GB GPUs (4 nodes × 4 GPUs). The training hour is 48. The specific hyparameters setting is detailed in Table 3

**Tab. 3: Training settings for agentic RL with a 4B SFT model.**

| Component | Configuration |
|---|---|
| Base model | Qwen3-4B-Instruct-2507 |
| SFT setup | Quantum Datasets (Jern et al., 2025) |
| SFT learning rate | $2 \times 10^{-5}$ |
| Rollout Num | 16 |
| Temperature | 0.7 |
| top_p | 0.8 |
| top_k | -1 |
| Token limits | 1024 (Prompt), 9216 (Response), 2048 (Observation), 8192 (Action) |
| Optimizer | AdamW (Loshchilov & Hutter, 2017) |
| Learning rate | $1 \times 10^{-6}$ |
| Epochs | 10 |
| Batch size | 128 |

## C  QUANTUM COMPUTING AND QUANTUM OPTIMIZATION

Quantum computing is an emerging computing paradigm that relies on the principles of quantum mechanics (Nielsen & Chuang, 2010). Some types of quantum computing hardware include superconducting circuits, photonic systems, trapped ions, spin qubits, and neutral atoms (Yang et al., 2023). While hardware often differs at a fundamental level, the most common abstraction to design quantum computing algorithms is the quantum circuit model, whose fundamental unit is a quantum bit. Whereas classical computing is based on discrete bits $0$ and $1$, quantum computing is based on quantum bits (qubits), which can be in superposition. A single qubit is defined as

$$|\varphi\rangle = \alpha|0\rangle + \beta|\varphi\rangle,$$

where $|0\rangle = \begin{bmatrix} 1 & 0 \end{bmatrix}^\top$ and $|0\rangle = \begin{bmatrix} 0 & 1 \end{bmatrix}^\top$ are column vectors and $\alpha, \beta \in \mathbb{C}$ so that $|\alpha|^2 + |\beta|^2 = 1$. A qubit is an element of a Hilbert space. Multi-qubit systems can be constructed using the tensor product of Hilbert spaces. Quantum computing is performed by applying quantum logic gates to the system of qubits. These gates are defined by unitary matrices $U$. Matrix $U$ is unitary if $UU^\dagger = U^\dagger U = 1$, where $U^\dagger$ is the conjugate transpose of $U$.

Considering the unitaries in this work, a special set of unitaries is the parametrized rotation gates, can be defined as

$$R_z(\theta) = \begin{bmatrix} e^{-i\theta/2} & 0 \\ 0 & e^{i\theta/2} \end{bmatrix}, \quad R_x(\theta) = \begin{bmatrix} \cos\frac{\theta}{2} & -i\sin\frac{\theta}{2} \\ -i\sin\frac{\theta}{2} & \cos\frac{\theta}{2} \end{bmatrix}, \quad R_y(\theta) = \begin{bmatrix} \cos\frac{\theta}{2} & -\sin\frac{\theta}{2} \\ \sin\frac{\theta}{2} & \cos\frac{\theta}{2} \end{bmatrix}.$$

These gates are unitary for every $\theta \in [0, 2\pi]$. Using these gates and other standard gates such as Hadamard, CNOT, and CZ gates, we can construct a vast class of parametrized quantum circuits whose elements we denote as $U(\theta)$, where $\theta = (\theta_1, \ldots, \theta_m)$ is a parameter vector. These types of parametrized gates serve as the basis for quantum optimization routines, which form the core set of circuits used as training data in this work. Next, we discuss quantum optimization.

Quantum optimization (Abbas et al., 2024) is one of the most promising applications in quantum computing. While quantum optimization can be performed with specialized devices, such as quantum annealers (Rajak et al., 2022), which are particular instances of adiabatic quantum computers (Farhi et al., 2000), multiple promising algorithms enable us to optimize a specific class of functions on universal quantum computers using the quantum circuit model. The key idea is to express a given optimization problem in a binary optimization format, which can be a quadratic unconstrained binary optimization (QUBO) problem (Lucas, 2014) or a higher-order unconstrained binary optimization (HUBO) problem (Boros & Hammer, 2002). Let $n$ be a positive integer and $[n] := \{1, \ldots, n\}$ be an indexing set. Formally, QUBO problems are then defined as the following minimization problem

$$\arg\min_{x \in \{0,1\}^n} \sum_{i \in [n]} \alpha_i x_i + \sum_{i < j} \alpha_{i,j} x_i x_j, \tag{5}$$

and HUBO problems are defined as

$$\arg\min_{x \in \{0,1\}^n} \sum_{S \subset [n]} \alpha_S \prod_{i \in S} x_i, \tag{6}$$

where coefficients $\alpha_i \in \mathbb{R}$ for $i \in [n]$. The QUBO problem is a special case of the HUBO problem where the variable interactions are limited to two. By performing a variable rewriting process

$$x_i \mapsto \frac{1}{2}(1 + z_i),$$

where $z_i \in \{-1, 1\}$ is a spin variable, we obtain the equivalent optimization formulations in terms of spin variables. By noting that the eigenvalues for the Pauli-Z operator

$$\sigma^z = \begin{bmatrix} 1 & 0 \\ 0 & -1 \end{bmatrix}$$

are 1 and $-1$, we can further map the spin variable formulation to the Hamiltonian

$$\sum_{S \subset [n]} \alpha_S \prod_{i \in S} \sigma_i^z,$$

where $\sigma_i^z$ is the Pauli-Z operator acting on qubit $i$. This rewriting process essentially translates the original binary optimization problem into an eigenvalue minimization problem for the Hamiltonian matrices that are the result of the unconstrained problems. The standard form of quantum optimization is based on the idea that we can prepare a state that enables us to measure a sufficiently low expectation value for a Hamiltonian, which depends on the QUBO or HUBO problem. The preparation of this state is done with a parametrized unitary $U(\theta)$. The goal is to estimate the gradient for the following function

$$f(\theta) := \langle 0 | U(\theta) H U^\dagger(\theta) | 0 \rangle, \tag{7}$$

which maps parameter vectors $\theta$ to expectation values of Hamiltonian $H$. For a fixed parameter vector $\theta$, $f(\theta)$ can be estimated with a quantum computer. By minimizing $f(\theta)$ with a suitable classical optimization algorithm, we are likely to prepare a state such that when we measure $\langle 0 | U(\theta)$ in the computational basis, the bitstring with the highest probability is a solution to the optimization problem.

The standard methods for solving quantum optimization problems on universal quantum computers include the Quantum Approximate Optimization Algorithm (QAOA) (Farhi et al., 2014), Variational Quantum Eigensolver (VQE) (Peruzzo et al., 2014), and adaptive VQE (Grimsley et al., 2019). Considering QAOA, the method requires preparing a special parameterized ansatz that consists of a circuit built based on the cost Hamiltonian that describes the optimization problem. The second part of the parametrized circuit is a mixer Hamiltonian, which is often a simple layer of parametrized $R_x$ rotation gates that act on every qubit in the system. Then, the expectation value for the cost Hamiltonian $H$ is measured as in Equation 7. By tuning the parameters with classical optimization methods, the goal is to minimize the expectation value. The VQE algorithm is similar except that the ansatz structure does not depend on the cost Hamiltonian, which enables the usage of either more expressive ansatzes or ansatzes that are hardware efficient. Otherwise, VQE is similarly a hybrid quantum-classical algorithm. Finally, adaptive VQE employs a method where the user defines a parametrized gate pool, where the algorithm picks gates and positions them in the circuit. Then, it evaluates the gradient and chooses the gate that performs the best. This leads to circuit structures that are problem-specific but highly adapted.

# D  TRAINING DATASET

## D.1  OFFLINE DATA COLLECTION

Effective verification requires access to accurate ground-truth metadata for each quantum optimization problem. Before training, we collect this metadata offline using the algorithms described in (Jern et al., 2025). For each optimization problem, we obtain (i) the cost Hamiltonian of each specific optimization problem, (ii) the parameterized solution (ground-truth) circuits, and (iii) the largest/smallest eigenvalues $E_{\max}$ and $E_{\min}$ for the ground-truth circuits with respect to the cost Hamiltonian. We feed this metadata into the quantum agent, which uses it to evaluate the OpenQASM code proposed by the language model.

## D.2  QUANTUM OPTIMIZATION PROBLEMS

### D.2.1  CONNECTED COMPONENT FOR A NODE

The problem of finding a connected component for a fixed node means finding a subgraph $G_s$ of graph $G$ such that $v_{\text{fix}} \in G_s$ and $G_s$ is a connected graph. A graph is connected if a path connects every two nodes in the graph. This formulation is based on two constraints. Let $V$ be the set of nodes in graph $G$ and let $x_v$ be the binary variable for each $v \in V$ indicating if the node belongs to the connected component or not. The first constraint is a so-called adjacency constraint term:

$$\sum_{v \in V} \left( |N(v)| \cdot x_v - \sum_{u \in N(v)} x_u \right)^2,$$

where $N(v)$ is the set of adjacent nodes to node $v$ and $|N(v)|$ is the size of this set. This constraint encodes the fact that if $x_v$ is activated, then we have to activate every variable linked to its neighbors, making the graph connected. Additionally, we include a regulation term

$$\sum_{v \in V} x_v,$$

which encodes the fact that we should not activate unnecessary variables, especially we should not activate every variable in the graph if they are not in the same connected component. Before solving the problem, we set $x_{v_{\text{fix}}} = 1$. This problem is not NP-hard, but it can be easily encoded as a QUBO and is a common graph optimization primitive. Thus, it provides a good example to be included in the training data.

### D.2.2  COMMUNITY DETECTION

The community detection problem seeks a partition $P$ of a graph $G$ so that the density of the edges within the partitions in $P$ is higher than the density of edges between them. The quality of the partitioning, i.e., communities, is often measured with modularity Clauset et al. (2004). The modularity-based community detection has a straightforward formulation in terms of QUBO optimization problems (Negre et al., 2020) if we consider dividing the graph into two communities.

Assume a weighted graph $G = (V, E)$ given as an adjacency matrix $A$, where $A_{ij}$ is either 0 if there is no edge between nodes $i \in V$ and $j \in V$, or $w_{ij}$, which is the weight for edge $ij \in E$. Define a node degree $d_i = \sum_j A_{ij}$ and collect the degree sequence to a vector $d = (d_1, \ldots, d_n)$, where $n = |V|$. Following (Negre et al., 2020), the modularity measure is defined as

$$B = A - \frac{dd^\top}{2m},$$

where $m = \sum_{ij} A_{ij}$. Then, we fix $K$ as the number of communities and define a set of binary variables $x_{v,k}$ for each node $v \in V$ and $1 \le k \le K$ indicating to which partition the node belongs. The QUBO objective that aims to maximize modularity is the following constraint

$$-\frac{1}{2m} \sum_{i,j \in V} \sum_{k=1}^{K} \left( A_{ij} - \frac{d_i d_j}{2m} \right) x_{i,k} x_{j,k}.$$

Note that the product $x_{i,k}x_{j,k}$ works as an indicator function: $i$ and $j$ are in the same community $k$ if and only if $x_{i,k}x_{j,k} = 1$. Moreover, we ensure that every node belongs to only a single community, which can be achieved with the following one-hot encoding

$$P \sum_{i \in V} \left(1 - \sum_{k=1}^{K} x_{i,k}\right)^2,$$

where the penalty factor $P$ should be sufficiently large. The problem is proved to be NP-hard Fortunato & Hric (2016).

### D.2.3   K-SIZED CLIQUE

Lucas (2014) presented the QUBO formulation for finding $k$-sized clique. The problem is to return a complete subgraph of size $k$ from a given graph $G$. The decision problem is NP-complete (Karp, 1972). The QUBO formulation for this problem is as follows. The first constraint encodes that we choose $k$ vertices

$$A \left(k - \sum_{v \in V} x_v\right)^2$$

and the second constraint encodes that we have to have $k(k-1)/2$ edges between the vertices

$$B \left(\frac{k(k-1)}{2} - \sum_{ij \in E} x_i x_j\right)^2.$$

The number of $k(k-1)/2$ edges characterizes a complete graph. Then, $A, B > 0$ are chosen so that $A > kB$.

### D.2.4   GRAPH ISOMORPHISM

Graph isomorphism between graphs $G_1$ and $G_2$ seeks a bijective mapping $f: V_1 \rightarrow V_2$ between the vertex sets of graphs $G_1$ and $G_2$ such that whenever $(v_1, v_2) \in E_1$ is an edge in graph $G_1$, then $(f(v_1), f(v_2)) \in E_2$ is an edge in graph $E_2$. Lucas (2014) describes the standard QUBO formulation for finding graph isomorphism with QUBO formulation, but formulations also exist for adiabatic quantum computers (Gaitan & Clark, 2014) and boson samplers (Brádler et al., 2021). Assuming that $u, v \in V_1$ and $i, j \in V_2$ are nodes, the first constraint is expressed as

$$A \sum_{v \in V_1} \left(1 - \sum_{i \in V_2} x_{v,i}\right)^2 + A \sum_{i \in V_2} \left(1 - \sum_{v \in V_1} x_{v,i}\right)^2,$$

which encodes the fact that there has to be a bijective mapping between the vertices. The second constraint encodes the fact that the bijective mapping has to respect edges

$$B \sum_{ij \notin E_1} \sum_{vu \in E_2} x_{v,i} x_{u,j} + B \sum_{ij \in E_1} \sum_{vu \notin E_2} x_{v,i} x_{u,j}.$$

It suffices to assume that $A, B > 0$.

### D.2.5   GRAPH COLORING

Assuming that $n$ colors and a graph $G$ are given, the graph coloring problem seeks a solution to the problem if the $n$ colors can be assigned to the vertices of $G$ so that no edge connects two vertices of the same color. The problem is known to be NP-complete Karp (1972). Lucas (2014) again presents the following formulation. Let $x_{v,i}$ be the binary variable indicating if the node $v$ should be colored with color $i$. The first constraint is the standard one-hot encoding, which states that every node should have one color

$$\sum_{v \in V} \left(1 - \sum_{i=1}^{n} x_{v,i}\right)^2.$$

The second constraint penalizes those cases when an edge connects two vertices with the same color

$$\sum_{uv \in E} \sum_{i=1}^{n} x_{u,i} x_{v,i}.$$

### D.2.6 Traveling salesman

The traveling salesman problem is one of the most studied optimization problems on a graph, where starting from a given node, the goal is to find a path in the weighted graph that visits every node in the graph exactly once. Lucas (2014) presents the following formulation with first constraint as

$$H_A = A \sum_{v=1}^{n} (1 - \sum_{j=1}^{N} x_{v,j}) + A \sum_{j=1}^{N} (1 - \sum_{v=1}^{n} x_{v,j})^2 + \sum_{(uv) \notin E} \sum_{j=1}^{N} x_{u,j} x_{v,j+1}$$

and the second constraint as

$$H_B = B \sum_{(uv) \in E} w_{uv} \sum_{j=1}^{N} x_{u,j} x_{v,j+1}$$

The decision problem is NP-complete (Karp, 1972). The penalizing terms can be chosen as $0 < B \max w_{uv} < A$ (Lucas, 2014).

### D.2.7 Weighted minimum maximal matching

A matching in a graph $G$ is a subset of its edges such that no two edges share the same vertex. Finding a matching is not generally NP-hard (Edmonds, 1965a;b) without additional constraints requiring minimality over the selected edges (Lucas, 2014). One of such formulations is to find a maximal matching on a weighted graph with the minimum cost Cook & Rohe (1999). A maximal matching is a solution where, if any edge not yet in the matching is included, the resulting subset of edges would no longer form a matching. Jern et al. (2025) considered this problem as a special instance of the exact set cover, where the edges are identified with two-element sets. This way, the exact set cover formula in (Lucas, 2014) can be used, simplifying the formulation. The first constraint enforces that for every node, exactly one edge is activated:

$$A \sum_{v \in V} \left( 1 - \sum_{e \in N(v)} x_e \right)^2$$

where $A = |V| + 1$. The second constraint encodes the fact that we want to minimize the weight of this matching

$$\sum_{e \in E} w_e.$$

### D.2.8 Vertex and edge covers

The vertex/edge cover problem seeks the smallest set of vertices/edges in a graph $G$ such that every edge/vertex has at least one vertex/edge in this set. The QUBO formulation (Lucas, 2014) for the vertex cover problem consists of two constraints

$$A \sum_{uv \in E} (1 - x_u)(1 - x_v)$$

and

$$B \sum_{v \in V} x_v,$$

where we choose $B < A$. The first constraint encodes the fact that every edge is connected to at least one vertex that is part of the cover. The second constraint aims to minimize the number of vertices in the cover. The decision problem is NP-complete (Karp, 1972). QAOA was previously benchmarked on a special version of this problem (Cook et al., 2019).

The edge cover admits a simple higher-order unconstrained binary optimization formulation as follows (Jern et al., 2025; Angara et al., 2025). The first constraint encodes that every vertex should be connected to at least one edge that is part of the cover:

$$A \sum_{v \in V} \prod_{e \in N(v)} (1 - x_e).$$

The previous constraint is a higher-order polynomial, which can be alternatively written as a quadratic polynomial using slack variables. The second constraint aims to minimize the size of the cover:

$$B \sum_{e \in E} x_e.$$

Again, we choose $B < A$.

### D.2.9 MAXFLOW AND MINCUT

A flow network is a directed graph with designated source and sink nodes, where each edge is assigned a non-negative capacity. The network contains no self-loops. A flow is given by a function $f \colon E \to \mathbb{R}$ that assigns a real value $f(u, v)$ for each edge $(u, v) \in E$ representing the amount of flow. A maximum flow problem is to find a viable flow from the source to the sink through the flow network, obtaining the maximum flow rate. Krauss et al. (2020) developed the QUBO formulation for the MaxFlow problem. In the same work, the authors also developed a QUBO formulation for the MinCut problem.

MaxFlow can first be presented as a quadratic unconstrained integer optimization problem. The first constraint encodes that for each edge, the input and output flow are equal:

$$\sum_{v \in V} \left( \sum_{e \in N^-(v)} z_e - \sum_{e \in N^+(v)} z_e \right)^2,$$

where $N^-(v)$ is the set of edges leaving node $v$, $N^+(v)$ is the set of edges coming to node $v$, and $z_e$ is the capacity of the edge, represented as an integer variable. Simultaneously, we want to maximize the flow, meaning we want to minimize

$$-A \sum_{e \in N^+(v)} z_e,$$

where $A > 0$ is a positive constant. Next, the encoding can employ the standard mechanism from (Lucas, 2014) to encode the integer variables as binary variables.

Due to the MaxFlow MinCut theorem, we automatically obtain a formulation for the MinCut problem as well. MinCut can be presented as

$$A(-x_s + x_s x_t) + \sum_{ij \in E} \alpha_{i,j}(x_i - x_i x_j),$$

where $x_s$ is the variable for the source node, $x_t$ is the variable for the target node, and $\alpha_{i,j}$ is the capacity or the weight. The part $x_i - x_i x_j$ evaluates 1 if $x_i = 1$ and $x_j = 0$, indicating that the edge is in the cut. We choose that $A > \sum_{ij \in E} \alpha_{ij}$.

## E COMPARISON WITH RANDOM BASELINE.

One of the most common parameter initialization methods for hybrid quantum-classical algorithms is still to use random parameters. Thus, it is crucial to understand if initial parameters in the QUASAR-generated QASMs outperform the randomized baseline. We evaluate this regarding two metrics: JS-divergence and expectation values. For every QUASAR-generated QASM, we computed the expectation value and JS-divergence with respect to the Hamiltonian and the ground truth. We parametrized the QUASAR-generated QASM and initialized 100 QASMs with uniformly randomly sampled parameters from the interval $(-\pi, \pi]$. For these QASMs, we computed the JS-diverge and expectation values. By comparing these metrics, we can see that both the distributions and expectation values from the QUASAR-generated QASMs are closer to the optimal than random parameter initializations on average. The results are collected in Table 4.

## F LIMITATIONS

We performed an experimental comparison between WarmStartQAOAOptimizer (Community, 2025) introduced in (Egger et al., 2021) to understand if QUASAR would work as a warm-start

Tab. 4: Comparison of QUASAR and random baseline metrics. Both JS-divergence and min-max-scaled expectation values are in $[0, 1]$, so that lower is better.

| Metric | Value |
|---|---|
| QUASAR JS-divergence | 0.79 |
| Random JS-divergence | 0.95 |
| QUASAR Expectation Value | 0.16 |
| Random Expectation Value | 0.36 |

method for quantum optimization. The WarmStartQAOAOptimizer utilizes a presolver, which solves a relaxed continuous variable version of the QUBO problem and then initializes the QAOA initial state and mixer accordingly. In our implementation, we utilized Gurobi, a high-performance industry-level solver. For the 564 syntactically correct QASMs in the test dataset (580 test cases in total), the average $f_{\max}^{\min}$ value for the WarmStartQAOAOptimizer is $0.007$, which necessarily indicates that the presolver was capable of solving the problems optimally on average and preparing a state and mixer that encoded the correct solution. The corresponding value for QUASAR was $0.1600$, which was also presented in Table 4. This result indicates that QUASAR is still limited as a warm-start method compared to the state-of-the-art rule-based WarmStartQAOAOptimizer.

On the other hand, WarmStartQAOAOptimizer works best with QUBO problems. Thus, QUASAR might be a viable method to warm-start more complex problems, such as HUBO problems, where WarmStartQAOAOptimizer's performance seemed to decrease. The method can be efficiently adapted to HUBO problems by simply optimizing the problem, which excludes the higher-order terms. By computing the values for those 60 HUBO problems in the test data set, which compiled correctly (62 HUBO test cases in total), the corresponding values are $0.0656$ for WarmStartQAOAOptimizer and $0.2356$ for QUASAR. While WarmStartQAOAOptimizer performance was an order of magnitude worse on these problems, it still outperformed QUASAR. To address these limitations, the training dataset could be extended with QASMs that the WarmStartQAOAOptimizer prepares for the QASMs in the training dataset.

