# OpenReview forum: "QUASAR: Quantum Assembly Code Generation Using Tool-Augmented LLMs via Agentic RL"
_ICLR.cc/2026/Conference — ICLR 2026 Conference Withdrawn Submission_

### Official Review · Reviewer_GbsY · 2025-10-24

**Soundness:** 2
**Presentation:** 2
**Contribution:** 2
**Rating:** 2
**Confidence:** 4

**Summary:**

The paper introduces QUASAR, an agentic reinforcement learning (RL) framework that augments large language models (LLMs) with external quantum simulators for quantum assembly code generation. The goal is to improve the syntactic and semantic correctness of OpenQASM 3.0 quantum circuits. QUASAR integrates supervised fine-tuning with a four-level hierarchical reward mechanism that incorporates syntactic validity, distributional alignment (via Jensen–Shannon distance), expectation-value alignment, and optimization progress. The approach is evaluated by augmenting a 4B-parameter LLM and compared with GPT-4o, GPT-5, DeepSeek-V3, and RL-only or SFT-only baselines. Results show high syntactic validity (99.31% Pass@1) and improved circuit quality on several quantum optimization benchmarks.

**Strengths:**

1.	The topic is timely and relevant, addressing quantum circuit generation using tool-augmented LLMs.
2.	The hierarchical reward mechanism is conceptually well motivated.
3.	The evaluation covers both syntactic and semantic metrics with clear quantitative reporting.
4.	The experimental setup includes multiple baselines and ablation studies.

**Weaknesses:**

1.	In Section 4.2 and Figure 3, the hierarchical reward mechanism, while interesting, lacks theoretical grounding or ablation analysis showing why the chosen four components (syntax, entropy, expectation value, optimization) are optimal. Other plausible metrics could exist, but justification is not provided.
2.	The comparison with GPT-4o and GPT-5 in Table 1 does not constitute a fair baseline against a fine-tuned model. The paper should clarify hyperparameter settings, prompt design, and reproducibility details for these baselines.
3.	Figure 5 reports ΔE distributions but omits units and normalization conventions. Without specifying whether values correspond to expectation differences or normalized eigenvalue gaps, it is difficult to interpret it.
4.	The Agentic RL in Section 3.3 largely reiterates standard GRPO methods (Shao et al., 2024) without adaptation to quantum contexts. The contribution seems incremental, since it applies an existing RL algorithm to a new domain with minimal innovation.
5.	The quantum verification pipeline in Section 4.1 is described only superficially. Implementation details of the “Quantum Tool Server” and simulation fidelity are missing. It is unclear whether noise, decoherence, or realistic hardware constraints were modeled.
6.	The reward normalization in Eq. (3) and Eq. (4) assumes bounded eigenvalues, but many Hamiltonians used in QAOA/VQE have variable scaling. This could bias the reward and affect convergence; no normalization consistency checks are discussed.
7.	Section 2 (Related Work) misses recent key works on quantum circuit compilation via differentiable programming and symbolic optimization. The related work is dominated by LLM-based citations and omits competing non-LLM approaches.
8.	Figure 2 and accompanying description do not specify the optimization problem (Hamiltonian) associated with the illustrated ansatz. It would be recommended to provide context, to improve clarity.
9.	The evaluation metrics in Section 5.1 rely on Pass@K-style measures, which are adapted from code generation. These metrics may not align with physical correctness or execution fidelity on real quantum backends. Including hardware-executed validation would strengthen the paper.
10.	The presentation has recurring typographical and formatting errors (e.g., “desigin” in the introduction, inconsistent use of subscripts in formulas), which reduce readability. Figures also have low resolution.
11.	The hierarchical reward ablation in Table 2 suggests only marginal gains from additional reward components, implying that most improvements could stem from data scale or SFT pretraining rather than from RL itself.
12.	While it has been promised to release the code after acceptance, it would be preferable to make the code available during the review in the supplementary material.
13.	The paper’s novelty lies mostly in combining existing techniques (OpenQASM simulation, GRPO RL, and LLM fine-tuning) rather than introducing a fundamentally new algorithmic insight.

**Questions:**

1.	How sensitive is the performance to the relative weighting of the four reward components?
2.	What mechanisms prevent reward hacking when circuits add extraneous qubits?
3.	How does QUASAR scale beyond 9-qubit or 12-qubit benchmarks?
4.	Can the hierarchical reward framework be generalized to other DSLs beyond OpenQASM?

---

### Official Review · Reviewer_1ovZ · 2025-10-29

**Soundness:** 2
**Presentation:** 3
**Contribution:** 2
**Rating:** 4
**Confidence:** 4

**Summary:**

This paper proposes QUASAR, an agentic RL framework to fine-tune LLMs to generate OpenQASM 3.0 programs for quantum optimization tasks. The method augments a 4B SFT model with a tool-use loop that calls an external quantum simulator and optimizes the policy with GRPO using a four-level hierarchical reward: (i) syntactic validity; (ii) distributional alignment via Jensen–Shannon distance with a qubit-mismatch penalty; (iii) expectation-value proximity to the ground truth problem Hamiltonian; and (iv) optimization-progress that rewards fewer classical optimization steps and better final value. The pipeline improves both syntax and semantics over SFT and strong prompting baselines on a dataset of graph-based optimization instances. Ablations suggest the distributional alignment term is the dominant driver, with expectation-value and optimization-progress giving complementary benefits.

**Strengths:**

- The reward shaping is well-designed. Clear decomposition into a hierarchy of four levels: syntax, distributional alignment, expectation value, and optimization progress. The qubit-mismatch penalty addresses a common failure mode of wrong wire counts. Ablations reinforces the effectiveness of RE term.
- Realistic training stack and reproducible high-level settings.

**Weaknesses:**

- **Overclaimed scope**. The title claims to be "quantum assembly code generation", and the introduction targets at general quantum circuits. However, all tasks, rewards, and metrics presuppose Hamiltonians + parameterized ansatzes. Nothing addresses general OpenQASM programs (e.g., QFT/PE, arithmetic, mid-circuit measurements, control flow, etc.). As far as I can see, the rewards and metrics cannot be adopted directly to universal quantum circuits, which limits the usage of this framework.
- **Limited conceptual novelty**.  The framework largely repackages a common agentic RL template. The quantum parts are well-crafted instantiations rather than new principles.
- **Gains over SFT are modest**. QUASAR’s main gains are semantic but the margins over SFT are somewhat incremental. An analysis of marginal improvement vs. extra compute would strengthen the case.
- **Threshold choices & metric redundancy (minor).** The SREV tolerance $|E(C)−E^\star∣\le 0.2$ is not justified. Sensitivity to this threshold should be reported. HQCR is defined as RE within 0.1, which is not necessary as a metric given RE in my opinion.
- **Lack of open-sourced code**. The abstract claims to provide training code at GitHub, but the linked repository is empty. Also an unanonymous link violates the double-blind review policy, which in principle should be desk-rejected.

**Questions:**

See Weaknesses.

---

### Official Review · Reviewer_nPXL · 2025-10-31

**Soundness:** 3
**Presentation:** 3
**Contribution:** 3
**Rating:** 4
**Confidence:** 4

**Summary:**

This paper proposes QUASAR, an agentic reinforcement learning framework for post-training large language models to generate parameterized quantum circuits in OpenQASM 3.0. It also introduced 4 hierarchical reward mechanism to enhance the effectiveness of the RL training process.

**Strengths:**

- Improves LLMs’ proficiency in PQCs generation.
- Well-structured and easy to follow
- A clear summary of quantum optimization problems.

**Weaknesses:**

- The motivation is not accurate
- Gap between reward mechanism and evaluation metrics

**Questions:**

- What is QUASAR's motivation? QUASAR look more like to enhance LLM generate better PQC structure and initial parameters, rather than generate quantum circuit. The distinction between these goals should be made clearer.
- What is the calculate process of the expectation-value reward? In Section 4.2 the expectation-value reward is calculated by the distance between the eigenvalues of the generated circuit and the ground truth circuit. However, in section 4.2.3 this was calculated by the problem specific cost Hamiltonian.
- Why use JS divergence as a reward. The JS divergence reward encourages the model to generate circuits whose unitaries closely match those of the ground-truth circuits. In contrast, the expectation-value reward and optimization-step reward aim to produce circuits that better approximate the target Hamiltonian. However, there remains an inherent gap between the dataset circuits and the ideal Hamiltonian solution. Therefore, the three rewards are not fully consistent in optimization direction. As the number of qubits increases, the JS divergence metric loses accuracy in capturing the difference between the two distributions, making it a less effective reward for high-dimensional quantum systems.
- Why SREV is used as the evaluation metric instead of directly using the expectation value percentage? Since the expectation value measures how closely the parameterized quantum circuit (PQC) approximates the target Hamiltonian, it is unclear why SREV was chosen as the primary indicator. According to the paper, SREV appears to better capture the approximation degree between the generated PQC and the target Hamiltonian. However, the experimental results show that increasing the expectation-value reward actually decreases SREV performance, which is counterintuitive. A detailed explanation of this discrepancy and the rationale behind selecting SREV over direct expectation-value measures would greatly improve the clarity of the evaluation section.
- How the training and test datasets are partitioned.
- How the prompts are constructed. It's better to give an example.

---

### Official Review · Reviewer_Evyu · 2025-11-01

**Soundness:** 2
**Presentation:** 2
**Contribution:** 2
**Rating:** 2
**Confidence:** 3

**Summary:**

The authors attempt to build quantum circuits efficiently with a specially trained LLM. In the measures defined by the authors, it outperforms unoptimized LLMs like GPT-5.

**Strengths:**

The application area of the paper is certainly novel. It indicates that superior performance can be obtained from an LLM even on tasks that are not well-suited if it is refined further.

**Weaknesses:**

The central idea, while imaginative, is not sufficiently grounded in the technical realities of quantum computing. Key challenges — such as the exponential scaling of gate requirements with qubit count — are not adequately addressed. Moreover, the paper provides limited information about the scale of the experimental setups, leaving it unclear how complex or realistic the tested instances are.

**Questions:**

- Could the authors clarify the size of the experiments, in terms of qubit count and circuit depth?
- Do you observe any changes in code generation performance as problem size increases?
- Are there notable differences in quality when generating dense versus sparse circuits?

**Details Of Ethics Concerns:**

Ethics statement is evidently wrong, LLMs were used for generating the results.

---

### Note · Authors · 2025-11-12

I have read and agree with the venue's withdrawal policy on behalf of myself and my co-authors.